

# An optimized deep learning approach for suicide detection through Arabic tweets

Nadiah A. Baghdadi[1], Amer Malki[2], Hossam Magdy Balaha[3], Yousry AbdulAzeem[4], Mahmoud Badawy[3] and Mostafa Elhosseini[2,3]

[1] Nursing Management and Education Department, College of Nursing, Princess Nourah bint Abdulrahman University, Riyadh, Saudi Arabia
[2] College of Computer Science and Engineering, Taibah University, Yanbu, Saudi Arabia
[3] Computers and Control Systems Engineering Department, Faculty of Engineering, Mansoura University, Mansoura, Egypt
[4] Computer Engineering Department, Misr Higher Institute for Engineering and Technology, Mansoura, Egypt

Corresponding author
Mahmoud Badawy,
engbadawy@mans.edu.eg

## ABSTRACT

Many people worldwide suffer from mental illnesses such as major depressive disorder (MDD), which affect their thoughts, behavior, and quality of life. Suicide is regarded as the second leading cause of death among teenagers when treatment is not received. Twitter is a platform for expressing their emotions and thoughts about many subjects. Many studies, including this one, suggest using social media data to track depression and other mental illnesses. Even though Arabic is widely spoken and has a complex syntax, depressive detection methods have not been applied to the language. The Arabic tweets dataset should be scraped and annotated first. Then, a complete framework for categorizing tweet inputs into two classes (such as Normal or Suicide) is suggested in this study. The article also proposes an Arabic tweet preprocessing algorithm that contrasts lemmatization, stemming, and various lexical analysis methods. Experiments are conducted using Twitter data scraped from the Internet. Five different annotators have annotated the data. Performance metrics are reported on the suggested dataset using the latest Bidirectional Encoder Representations from Transformers (BERT) and Universal Sentence Encoder (USE) models. The measured performance metrics are balanced accuracy, specificity, F1-score, IoU, ROC, Youden Index, NPV, and weighted sum metric (WSM). Regarding USE models, the best-weighted sum metric (WSM) is 80.2%, and with regards to Arabic BERT models, the best WSM is 95.26%.

## INTRODUCTION

Millions of individuals worldwide suffer from various mental disorders. These disorders impact their thoughts and conduct, and their quality of living (*Kessler et al., 2017*; *Mathers & Loncar, 2006*). One of the most common mental illnesses globally is major depressive disorder (MDD), also known as depression. It has a significant influence on an individual's daily activities. It is linked to functional impairment, which is expensive to society. More than 300 million individuals are impacted globally, accounting for 4.4% of the global population, and the number is growing every day (*World Health Organization (WHO),*

*2021*). According to estimates, unipolar depression will contribute to the second-largest portion of the global illness burden by 2030. It is important to differentiate depression from the mood swings and transient emotions we experience every day (*Russell, 1980*). If it persists for longer than 2 weeks, it can become a serious health problem. It can impact both a person's mental and physical well-being. Thus, depressed people exhibit poor performance and behave unprofessionally at work and at home. If a depressed individual does not receive sufficient care, it might lead to suicide.

Every year, about 800,000 individuals commit suicide (*Safa, Bayat & Moghtader, 2022*). Among teenagers, it is the second leading cause of death. Depression affects millions of individuals worldwide, regardless of culture, gender, age, ethnicity, or economic condition. Symptoms of depression can be categorized psychologically, socially, and physically (*NHS, 2019*). The presence of all these symptoms is rare in depression patients, but they can predict the severity of the illness. Although a high percentage of depression is treated, over 70% of those affected do not receive therapy, which may cause their illness to progress (*Salas-Zárate et al., 2022*; *DBSA, 2022*). This emphasizes the importance of excellent mental health services and an innovative way to detect mental health disorders.

Traditional interactions between individuals and organizations have evolved. Meetings and conversations occur in online communities to exchange knowledge, obtain and provide assistance, and expand the business. Individuals use social media platforms such as Twitter and Facebook to express their emotions and thoughts about a wide range of topics (*Safa, Bayat & Moghtader, 2022*). An open and free communication channel such as social media facilitates problem resolution, and knowledge exchange (*Salas-Zárate et al., 2022*). The current status of society is deeply entwined with social networks. Last year, social media users climbed by around 424 million, bringing the overall number of users to 4.62 billion (*Kemp, 2022*). This equates to an annual increase of more than 10%. This evolution of social media enabled individuals to instantly express their feelings, ideas, emotions, and opinions. In the process, huge amounts of social data are generated, providing useful information about people's mental and everyday activities (*Wongkoblap, Vadillo & Curcin, 2022*). Mining and analyzing user data gives a unique chance to research human behavior in depth and their state of mental health and assist them in recovering.

Because depression influences human behavior, researchers devised a method for tracking depression and other mental diseases using social media data. The conventional study method for depression analysis relies on questionnaires, which involve participants providing subjective responses or comments. Social media data alone cannot provide people's true emotions since they differ from person to person and are hard to get. As a result, social media is frequently employed to detect stress, anxiety, and depression disorders. Research on psychology and human behavior can be conducted through social media sites like Twitter, and Facebook (*Mishra & Garg, 2018*; *Kim et al., 2020*). For example, studying tweets of individuals who have experienced severe depression has proven useful in predicting future depression in those individuals (*AlSagri & Ykhlef, 2020*; *Zafar & Chitnis, 2020*).

The analysis of social media data allows for the discreet detection of depressive symptoms before they proceed to more severe stages of depression. This enables the

recommendation of strategies for depression prevention and therapy in the early phases. Social media sites have been used to identify depression, with Twitter being the most commonly utilized (*Smys & Raj, 2021*; *Liu et al., 2022a*). Over 436 million monthly active Twitter users send more than 500 million Tweets each day, making Twitter the ninth most popular website on the Internet in terms of popularity (*Kemp, 2022*). Any registered user can publish his or her opinions in 140 characters at a time. Tweets are often made public and may be collected and analyzed using the Twitter API (*Liu et al., 2022b*). In addition, Twitter's API enables complicated searches such as retrieving every tweet regarding a specific topic (*Vanam & Jeberson, 2021*).

Speech and text analysis is carried out through natural language processing (NLP). During the development of computer technology, linguistic approaches evolved into computer algorithms. Natural language processing was originally used to examine classical tasks, such as structuring grammar in books. However, it has expanded to interpreting human viewpoints, including mail, online content, reviews, social networking postings, and media articles (*Zhang et al., 2022*). In many ways, NLP can process a variety of languages (*Islam et al., 2018*). For example, using sentiment analysis (SA), you can identify positive and negative aspects of a text or speech (*Bhushan & Sharma, 2021*). This method may be adapted to analyze a person's depression levels based on social media posts and negative sentiment ratings. Many prediction algorithms and optimization approaches exist for noticing patterns in data and forming insight based on those observations, such as deep learning (DL) and machine learning (ML). For example, when researchers use binary classification approaches for text analysis, they use traditional machine learning (ML) algorithms such as random forest (RF), support vector machines (SVMs), decision trees (DT), *etc.*

Understanding the available dataset can aid in determining whether machine learning or deep learning should be used to solve a particular problem. Machine learning is typically utilized when structured data are scarce. Most machine learning algorithms are built to train models using tabular data. Deep learning usually necessitates a significant amount of training data to ensure that the network, which may contain tens of millions of parameters, does not overfit the data. Hardware also plays an important role in choosing which AI method to use. For instance, machine learning algorithms use less computing power. Therefore, desktop computers are sufficient for training these types of algorithms. However, deep learning models require specialist hardware due to their high memory and computation requirements. To summarize, for deep learning to work, a tremendous amount of data must be collected, but minimal human intervention is required. Unfortunately, there is no cure for the shortage of large training datasets that transfer learning provides.However, the traditional ML technique's performance was restricted as the number of correlations rose dramatically due to the rise in data amount (*Rao et al., 2020*).

The use of deep learning (DL) in a variety of areas has been successfully demonstrated in recent years, including stock market forecasting (*Ni, Wang & Cheng, 2021*), traffic flow and accident risk prediction (*Essien et al., 2021*), and mental disease diagnosis (*Kim et al., 2020*). Deep learning has also been used to detect depression on social media, and the

results were significantly better than those achieved with traditional machine learning approaches. In addition, using Deep Learning models like RNNs and CNNs coupled with neural word embeddings has significantly improved textual data classification accuracy (*Sood et al., 2018*). This article presents a framework for analyzing texts (tweets) using six phases, starting with data acquisition and ending with classification, moving through four more steps. Finally, a trained and optimized model is used to classify the Arabic tweets into suicides or normal tweets. Accordingly, the following points outline the study's findings:

– Providing a comprehensive framework for classifying Arabic texts.
– Proposing a method of preprocessing tweets.
– Using the latest Arabic and universal models.
– Suggesting scraping and annotating the Arabic tweets dataset.
– Comparing different methods of lexical analysis, such as lemmatization and stemming.
– Reporting metrics of state-of-the-art performances from the Arabic tweet dataset.

What follows is a summary of the rest of the article: In "Related work", the related studies are presented. "Methodology" discusses the methodology in six phases and presents the suggested framework and algorithm. "Experiments and discussions" reports the experiments and their results and discussions. Finally, the article is concluded in "Conclusions", and the future work is presented therein.

## RELATED WORK

Using social media data to detect depression has been the subject of many research works. Some of these research techniques are discussed here. *Victor et al. (2019)* developed multi-modal neural networks for detecting depression. Automatic evaluation is conducted using data collected from 671 participants. To evaluate depression, participants completed the Patient Health Questionnaire (PHQ-9) (*Kroenke, Spitzer & Williams, 2001*). This model uses artificial intelligence and the naive Bayes classification method to enhance system accuracy. They achieved 70.88% accuracy. In a recommendation system designed by *Rosa et al. (2018)*, CNN was used to identify users who may be suffering from psychological problems, such as depression and stress. One hundred forty-six assessors participated in a web-based questionnaire to build the system to identify their emotional states. The system delivers comforting messages to the user through 360 predefined messages. *Wang, Niu & Yu (2019)* developed SentiDiff, a novel algorithm based on the correlation between tweets and retweets. Combining this model with a traditional neural network resulted in a 79% accuracy. Nonetheless, the algorithm only considered tweets with retweets attached and only analyzed text-based information.

With CNN, *Akhtar et al. (2019)* achieved an accuracy of 80.52% in a single analysis, focusing on rough and fine-grained emotions and sentiment. They built a multi-task ensemble framework for emotion, sentiment, and intensity prediction. The framework employed a multi-layer perceptron network that learns from three deep learning models (*i.e.*, CNN, LSTM, and GRU) assisted by a handcrafted feature vector. The multi-task ensemble network achieved an accuracy of 89.88%. *Suman et al. (2020)* analyzed tweets for

sadness using DL models combined with a cloud-based app. This study employed RoBERTa to classify tweets using the linked tweets. The study achieved a testing accuracy of 87.23%. *Shetty et al. (2020)* employed SA to analyze Twitter tweets. For DL, KAGGLE datasets were used, and LSTM was used to build deep networks. CNNs were later proposed in the classification phase for improved performance. To discover the intensity of the sentiments of extremism, *Asif et al. (2020)* employed multinomial naive Bayes and linear support vector classifier algorithms on social media textual data. They classified the incorporated views into four categories based on their level of extremism. During the COVID-19 epidemic, *Alabdulkreem (2021)* adopted ML to analyze Arab women's tweets to determine whether they had depression symptoms. From 10,000 tweets taken from 200 individuals, they used an RNN approach to predict depression. They achieved an accuracy of 72%.

According to *Chiu et al. (2021)*, using Instagram photos, text, and behavior characteristics of posts, a multi-model architecture is designed to predict depression based on the DL. They applied an aggregation method with a two-stage detection mechanism to recognize the hidden emotions of people. *Tommasel et al. (2021)* analyzed social media expression in Argentina as part of their DL approach. Time-series data were generated (using markers) and entered into a neural network during the COVID-19 pandemic to predict psychological health and feelings. In a study by *Jyothi Prasanth, Dhalia Sweetlin & Sruthi (2022)*, they developed a method to diagnose depression among Twitter users by reducing multiclassification rates. An improved method for classifying negative tweets is proposed. A glossary of emoticons is created, and genuinely negative tweets are detected. They achieved accuracies of 72%, 76%, and 90% for RNN, LSTM, and BiLSTM networks, respectively. *Zogan et al. (2022)* developed explainable Multi-aspect depression detection utilizing social media analysis to detect depressed people. It employs a hierarchical attention network to extract users' online activity and behavior characteristics. They also compared the proposed technique with both traditional and deep learning techniques. Their system achieved 89.5% accuracy. *Kour & Gupta (2022)* developed a unique way of predicting depression using raw Twitter datasets by combining CNN and biLSTM networks. To analyze longer text sequences, the proposed model employs biLSTM. *Nair et al. (2022)* employed machine learning techniques to predict early signs of depression. Classifiers such as naive Bayes, decision trees, SVM, and k-nearest neighbors are employed on big data extracted from social media. The decision trees classifier achieved the highest results with 97% accuracy. Using clinical interviews, *Park & Moon (2022)* built a multi-modal system to detect depression. They used BERT-CNN for natural language analysis and CNN-BiLSTM for voice signal processing. They proved that a fusion of text data and voice data improves detection accuracy.

*Safa, Bayat & Moghtader (2022)* proposed a multi-modal framework to predict depression symptoms from Twitter profiles. They applied lexicon analysis and NLP techniques to study the relationship between language and depression. In addition, they explored new features, such as bio-text, profile picture, and header image. To identify image content, a convolutional neural network-based automatic tagging system is used to tag both profile and header images. Their results showed that the correlation-based

method leads to better outcomes. *Lia et al. (2022)* used six generic machine learning algorithms to detect depression from Twitter's tweets. They used term frequency-inverse document frequency for feature extraction before applying classification algorithms. Among the used techniques, the support vector classifier outperforms all the other techniques with 79.90% accuracy. *Shankdhar, Mishra & Shukla (2022)* used natural language processing tools and deep learning algorithms to predict depression through users' tweets. First, they built a baseline model using LSTM and achieved an accuracy of 95.12%. Afterward, they enhanced the model using a hybrid BiLSTM+CNN model. BiLSTM layers extract superior sequence features from text data, whereas CNN improves classification with higher dimensional features. To identify depression early on, *Tong et al. (2022)* proposed a new classifier, namely, cost-sensitive boosting pruning trees. They used their classifier on five distinct datasets, including two publicly available Twitter depression detection datasets and three datasets from the UCI machine learning repository. The proposed classifier is compared to state-of-the-art Boosting algorithms to assess it. In addition, they merged the Discrete Adaboost structure with the pruning procedure (Adaboost+PT) as a comparison approach. The proposed classifier outperformed the previous depression detection methods in the two Twitter datasets.

Table 1 provides a brief comparison of the techniques under consideration. Language, data source, data size, and achieved accuracy are stated for each technique. As demonstrated by the comparison, there is a lack of depression detection approaches applied to the Arabic language, despite its widespread use and difficult syntax. There is a crucial need for an empirical quantitative framework for text classification and depression detection for Arabic.

## METHODOLOGY

The current study suggests a framework to perform text classification in six phases. It is summarized graphically in Fig. 1 and discussed in the following subsections. In practice, the Arabic tweet is classified using the optimized and trained model, and the output is generated to be "Suicide" or "Normal" as shown in Fig. 2. It tried to evaluate seven tweets that predicted six correctly and one incorrectly.

### Phase 1: data acquisition

The authors scraped the data from Twitter using the "Twint" Python package (*TWINT, 2022*). Twint is an advanced Twitter scraping and OSINT tool. The authors worked on scraping six keywords (with and without hashtags) that have the native meaning of the English word "Suicide". They are listed in Table 2. The authors tried to retrieve the tweets from "2017-01-01" to "2022-01-20" (*i.e.*, when the script was running). The authors retrieved 14,576 tweets concerning the search keywords and specified duration. Table 3 show five samples from the retrieved records.

### Phase 2: data pre-processing

With an initial investigation, after the scraping phase, multiple tweets are duplicated or meaningless (*i.e.*, containing only keywords, links, and/or emojis). This can be considered

**Table 1  A comparison between state-of-the-art depression detection techniques.**

| Paper | Year | Technique | Language | Data source | Data size | Accuracy (%) |
|---|---|---|---|---|---|---|
| Victor et al. (2019) | 2019 | BiLSTM | English | Questionnaire | 671 participants | 70.88 |
| Rosa et al. (2018) | 2019 | CNN+BiLSTM | Portuguese | Questionnaire | 146 participants | 90 |
| Wang, Niu & Yu (2019) | 2020 | Sentiment diffusion patterns | English | Twitter | 100,000 tweets | 79 |
| Akhtar et al. (2019) | 2020 | CNN+LSTM+GRU | English | EmoInt | 7,102 | 89.88 |
| | | | | EmoBank | 10,062 | |
| | | | | Facebook post | 2,895 | |
| | | | | SemEval-2016 | 28,632 | |
| Suman et al. (2020) | 2020 | DL | English | Twitter | 1,600,000 | 87.23 |
| Shetty et al. (2020) | 2020 | LSTM | English | Twitter | N/A | 70 |
| | | Model vector | | | | 76.69 |
| Alabdulkreem (2021) | 2021 | RNN-LSTM | Arabic | Twitter | 10,000 | 72 |
| Chiu et al. (2021) | 2021 | CNN for image | English | Instagram | 520 users | N/A |
| | | Word2vec for text | | | | |
| | | Handcrafted features for behavior | | | | |
| Tommasel et al. (2021) | 2021 | RNN | Spanish | Twitter | 150 million tweets | N/A |
| Jyothi Prasanth, Dhalia Sweetlin & Sruthi (2022) | 2022 | RNN | English | Twitter | 1,200 users | 72 |
| | | LSTM | | | | 76 |
| | | BiLSTM | | | | 90 |
| Zogan et al. (2022) | 2022 | CNN+RNN | English | Twitter | 4,208 users | 89.5 |
| Kour & Gupta (2022) | 2022 | CNN-biLSTM | English | Twitter | 1,402 depressed | 94.28 |
| | | | | | 300 million non-depressed | |
| Nair et al. (2022) | 2022 | Machine learning | English | Twitter | 10,000 tweets | 97 |
| Park & Moon (2022) | 2022 | BERT-CNN | English | DAIC-WOZ | N/A | N/A |
| Safa, Bayat & Moghtader (2022) | 2022 | Machine learning | English | Twitter | 11,890,632 tweets | 91 |
| Lia et al. (2022) | 2022 | Machine learning | English | Twitter | 1,600,000 | 79.9 |
| Shankdhar, Mishra & Shukla (2022) | 2022 | BiLSTM+CNN | English | Twitter | 12,274 tweets | 95.12 |
| Tong et al. (2022) | 2022 | Discrete Adaboost Cost-sensitive Boosting Pruning Trees | English | TTDD | 7,873 tweets | 88.39 |
| | | | | CLPsych 2015 | 1,746 tweets | 70.69 |
| | | | | LSVT | 128 | 85.72 |
| | | | | Statlog | 6,435 | 92.21 |
| | | | | Glass | 214 | 77.63 |

noise in the data; hence, a pre-processing phase is necessary for the current scenario. Different pre-processing steps are applied as shown in Fig. 3. The steps include: (1) the scraped tweets are read, (2) the duplicated tweets are eliminated, (3) the suggested Arabic
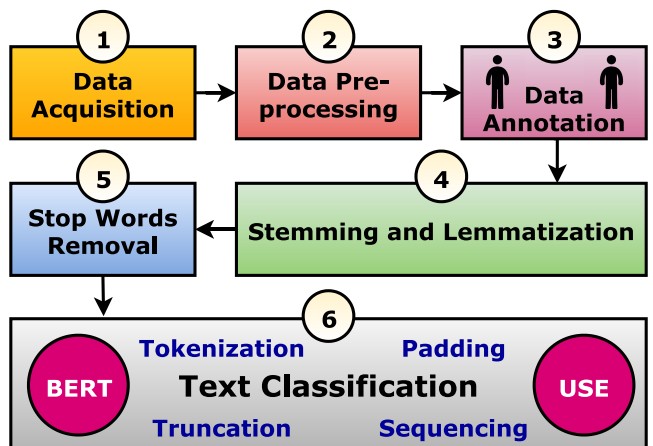

**Figure 1** The suggested text classification framework.

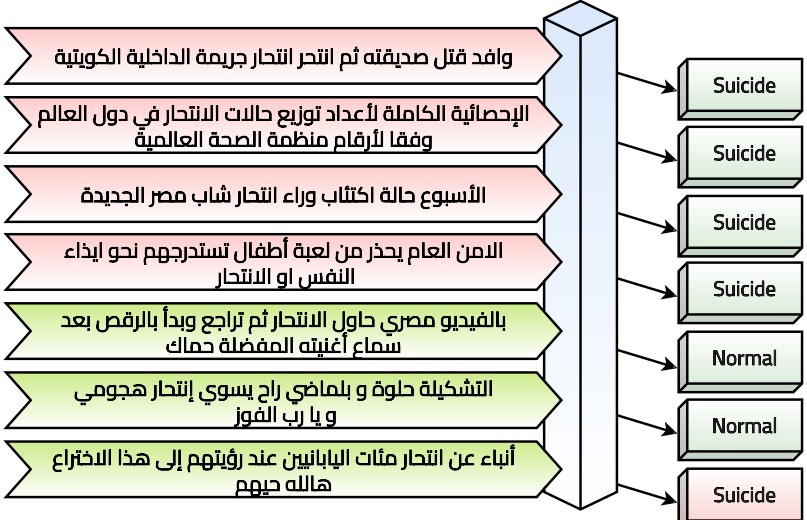

**Figure 2** System evaluation in practice.

**Table 2** The six scraped keywords (with and without hashtags).

| Sr. | Without hashtag | With hashtag |
|---|---|---|
| 1 | انتحار | انتحار # |
| 2 | الإنتحار | الإنتحار # |
| 3 | الانتحار | الانتحار # |
| 4 | إنتحار | إنتحار # |
| 5 | منتحر | منتحر # |
| 6 | سانتحر | سانتحر # |

RegEx pre-processing algorithm (discussed in the following paragraph) is applied, (4) the empty and duplicated tweets are eliminated, and (5) the tweets are sorted in ascending order concerning the create timestamp (*i.e.*, data, time, and timezone).

**Table 3  Five samples from the scraped tweets.**

| Sr. | Tweet |
|---|---|
| 1 | إنتحار سياسي |
| 2 | التشكيلة حلوة و بلماضي راح يسوي إنتحار هجومي و يا رب الفوز |
| 3 | محاولة انتحار سيا بعد انتقادات لفيلمها |
| 4 | انتحار شاب في الناصرية حي اور |
| 5 | انتحار شابة بسبب دواء ضد حب الشباب |

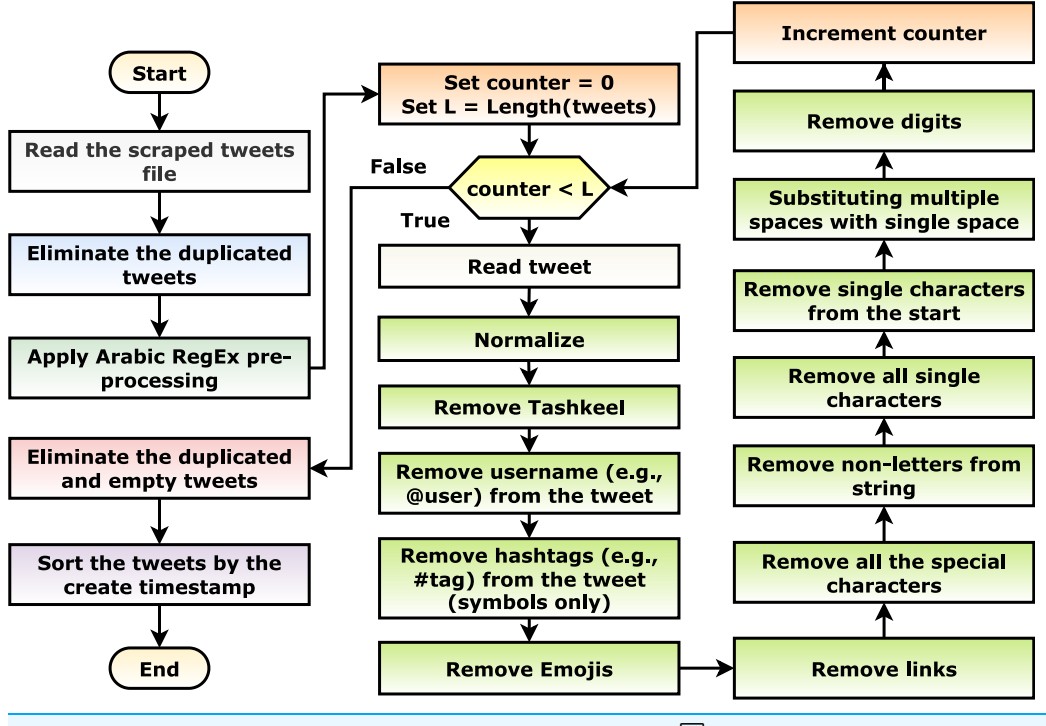

**Figure 3  The tweets pre-processing flowchart.**     

The suggested Arabic regular expression pre-processing algorithm iterates on each tweet and apply: (1) normalize the tweet by changing a set of characters to its origin character (Table 4), (2) remove the Tashkeel (*i.e.*, "Shadda", "Fatha", "Tanwin Fath", "Damma", "Tanwin Damm", "Kasra", "Tanwin Kasr", "Sukun", and "Tatwil/Kashida"), (3) remove the username (*e.g.*, @hossam) using the "`@[^\s]+`" regular expression, (4) remove the hashtag symbol (*i.e.*, "#"), (5) remove the emojis, (6) remove the links using the "`(?:\@|http?\://|https?\://|www)\S+`" and "`((www\.[^\s]+)|(https?://[^\s]+))`" regular expressions, (7) remove all of the special characters using the "`\W`" regular expression, (8) remove the non-letters using the "`[^0-9\u0600-\u06FF]+`" regular expression, (9) remove all of the single characters using the "`\s+[0-9\u0600-\u06FF]\s+`" regular expression, (10) remove the single characters from the beginning of the tweet using the "`\^[0-9\u0600-\u06FF]\s+`" regular expression, (11) substitute the

Table 4 The normalized characters.

| Sr. | From character(s) | To character |
|-----|-------------------|--------------|
| 1 | ا آ أ إ | ا |
| 2 | ي | ى |
| 3 | ة | ه |
| 4 | گ | ك |
| 5 | ؤ ئ | ء |

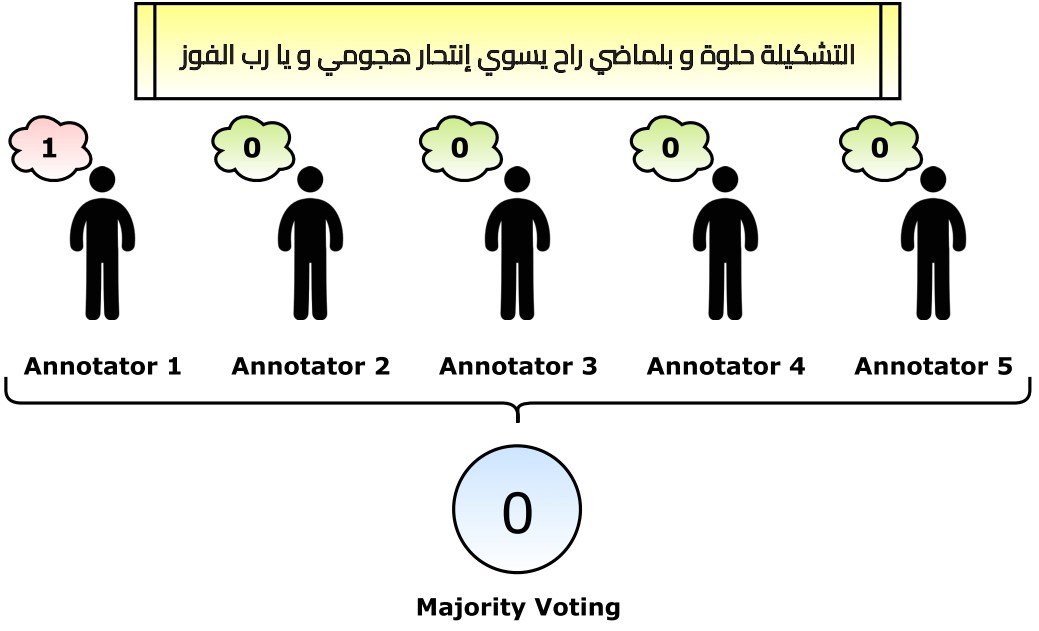

Figure 4 Graphical summarization of the voting process.

multiple white spaces with single ones using the "\s+" and "+" regular expressions, and finally, (12) remove the digits using the "[0-9]+" regular expression.

## Phase 3: data annotation

After applying the pre-processing steps, the number of remaining and unique tweets is 2,030. The next phase is to annotate the dataset. Five annotators helped in the annotation phase. Each annotator retrieved an empty sheet with the tweets only. He/she read the tweets one by one and marked "1" for the suicide tweets and "0" otherwise. The criteria for marking a suicide tweet is the tweet context. In other words, if the real contextual meaning of the tweet shows suicide, then it is marked by "1". The reason is that some tweets reflect some different meanings, such as sarcasm (*e.g.*, the fifth row in Table 3). After that, the majority voting algorithm is applied to gain the correct annotations. For example, if a tweet had [1, 0, 0, 1, 1], then it is considered a suicide tweet because the number of ones is more than zeros. Figure 4 reflects a graphical summarization of the voting process. After the annotation process, 1,074 tweets are categorized as "Normal" and 956 are categorized as

**Table 5 Six samples from the processed and annotated tweets.**

| Sr. | Tweet | Category |
|---|---|---|
| 1 | وافد قتل صديقته ثم انتحر انتحار جريمة الداخلية الكويتية | 1 |
| 2 | الإحصائية الكاملة لأعداد توزيع حالات الانتحار في دول العالم وفقا لأرقام منظمة الصحة العالمية | 1 |
| 3 | الأسبوع حالة اكتئاب وراء انتحار شاب مصر الجديدة | 1 |
| 4 | الامن العام يحذر من لعبة أطفال تستدرجهم نحو ايذاء النفس او الانتحار | 1 |
| 5 | بالفيديو مصري حاول الانتحار ثم تراجع وبدأ بالرقص بعد سماع أغنيته المفضلة حماك | 0 |
| 6 | أنباء عن انتحار مئات اليابانيين عند رؤيتهم إلى هذا الاختراع هالله حيهم | 0 |

**Table 6 A sample comparison between a stemmed and lemmatized tweet.**

| Type | Tweet |
|---|---|
| Before Normalization | شاب ينتحر بسبب ما فعلته مع أسرته ورسالته قبل الانتحار بدقائق أنا رايح لربنا يجيبلي حقي صور |
| After Normalization | شاب ينتحر بسبب ما فعلته مع اسرته ورسالته قبل الانتحار بدقاءق انا رايح لربنا يجيبلي حقي صور |
| Stemmed | شاب نحر سبب ما فعل مع اسر رسل قبل نحر دقءق انا ريح لرب جيبل حقي صور |
| Lemmatized | شاب انتحر سبب ما فعل مع اسره رسالة قبل انتحار بدقاءق انا رايح رب يجيبلي حق صور |

"Suicide". Samples from the annotated and processed tweets are shown in Table 5. In the future, to overcome this issue for Arabic tweets, automatic learning can be applied as follows: (1) the built system can accept the future tweets and predict them, (2) volunteer annotators can determine whether the system predicted correctly or wrongly, and (3) the new tweet can be used to update the model weights.

## Phase 4: stemming and lemmatization

The stemming and lemmatization processes are applied to the pre-processed and annotated tweets. The target of stemming is to remove (*i.e.*, stem) the last few characters of a certain word. It can lead to incorrect meanings and spelling. On the other hand, lemmatization considers the context and converts the word to its meaningful base form (*i.e.*, lemma). The Information Science Research Institute's (ISRI) Arabic Stemmer (*Taghva, Elkhoury & Coombs, 2005*) and Qalsadi Arabic Lemmatizer (*Zerrouki, 2012*) are used to perform that. A stemmed and lemmatized tweet is shown in Table 6.

## Phase 5: stop words removal

After the stemming and lemmatization phase, the stop words are removed from the tweets. The current study worked on 754 stop words in Arabic (*NLTK, 2022*). The process is to scan each token (*i.e.*, single word) in the tweet, if it belongs to the stop words, then it is eliminated. Examples of the used stop words are: "شرع" "كاد", and "مازال". In the current phase, there are six different datasets flavours to compare with. They are (1) the tweets before normalization, (2) the tweets after normalization, (3) the stemmed and normalized tweets, (4) the lemmatized and normalized tweets, (5) the filtered stemmed and normalized tweets, and (6) the filtered lemmatized and normalized tweets.

## Phase 6: classification

The data consists of tweets and the corresponding labels. A tweet vector can be though a distributed representation of that tweet and computed for every tweet in the dataset. The tweet vector and the corresponding label are used to train and validate the statistical classification model. The intuition is that the tweet vector captures the tweet semantics and hence can be effectively used for classification. Two alternatives, common for several text classification problems in NLP, can be used to obtain the tweets vectors. They are (1) word-based and (2) context-based representations. The word-based representation combines word embeddings of the tweet content words, and then the average of the word embeddings is calculated. The context-based representation can handle language models to generate vectors of the sentences. So, instead of learning vectors for individual words in the sentence as happens with the word-based representation, that latter representation computes a vector for sentences as a whole by considering the order of words and the set of co-occurring words.

GloVe and Word2Vec are examples of the first representation, while Embeddings from Language Models (ELMo), Neural-Net Language Model (NNLM), Bidirectional Encoder Representations from Transformers (BERT), and Universal Sentence Encoder (USE) are examples on the second representation. *Joshi et al. (2019)* recommend the latter approach than the first approach. The current study utilizes the USE and BERT with the Arabic flavors to perform the text classification task. The BERT uses the transformer architecture besides a set of different techniques to train a model, resulting in a state-of-the-art performance. The USE also uses the transformer architecture, not the recurrent neural networks (RNNs), similar to the ELMo approach.

For the USE, the universal-sentence-encoder-multilingual (*Google, 2019a*) and universal-sentence-encoder-multilingual-qa (*Google, 2019b*) are used. They support the Arabic language alongside another 15 languages. The used architecture is built as follows (1) the USE embedding layer, (2) a dense layer with 128 neurons and ReLU activation function, (3) another dense layer with 64 neurons and ReLU activation function, and (4) an output activation function with the SoftMax activation function. The embedding size is set to 256. The optimizer is set to Adam with a learning rate of 0.0001. The loss function is set to "Categorical Crossentropy".

For the BERT, the Arabic BERT versions: AraBERT (*Antoun, Baly & Hajj, 2020*) and AraELECTRA (*Antoun, Baly & Hajj, 2021*) are used. Five different flavors from them are used: "bert-base-arabertv2", "araelectra-base-generator", "bert-base-arabertv02-twitter", "bert-large-arabertv02-twitter", and "bert-large-arabertv2". Intuitively, using BERT models requires additional pre-processing as the model does not perform this automatically. Hence, the sentences are tokenized, truncated, padded, and converted to a sequence before being classified. The max sequence length is set to 100.

Table 7 summarizes the used models and their configurations. The last column defines a keyword to be used in the experiments instead of using the whole model name. To evaluate the model, different performance metrics are calculated (*Seliya, Khoshgoftaar & Van Hulse, 2009*; *Baghdadi et al., 2022b*, *2022a*): (1) accuracy (Eq. (1)), (2) balanced accuracy

**Table 7 Summary on the used models and the their configurations.**

| Approach | Model name | Size | Pre-process? | Experiment keyword |
|---|---|---|---|---|
| USE | universal-sentence-encoder-multilingual | 245.48 MB | No | USE-M |
| USE | universal-sentence-encoder-multilingual-qa | 310.68 MB | No | USE-MQA |
| USE | universal-sentence-encoder-multilingual (Tuned) | 245.48 MB | No | USE-M (Tuned) |
| USE | universal-sentence-encoder-multilingual-qa (Tuned) | 310.68 MB | No | USE-MQA (Tuned) |
| BERT | bert-base-arabertv2 | 543 MB | Yes | BERT-BV2 |
| BERT | bert-large-arabertv2 | 1.38 GB | Yes | BERT-LV2 |
| BERT | araelectra-base-generator | 227 MB | Yes | AraElectra |
| BERT | bert-base-arabertv02-twitter | 543 MB | Yes | BERT-BV02T |
| BERT | bert-large-arabertv02-twitter | 1.38 GB | Yes | BERT-LV02T |

(Eq. (2)), (3) precision (Eq. (3)), (4) specificity (Eq. (4)), (5) recall (*i.e.*, sensitivity) (Eq. (5)), (6) F1-score (Eq. (6)), (7) IoU (Eq. (7)), (8) ROC (Eq. (8)), (9) Youden Index (Eq. (9)), and (10) NPV (Eq. (10)). In addition to them, a weighted sum metric (WSM) is calculated mentioned metrics using Eq. (11).

$$\text{Accuracy} = \frac{TP + TN}{TP + TN + FP + FN} \tag{1}$$

$$\text{Balanced Accuracy} = 0.5 \times (\text{Recall} + \text{Specificity}) \tag{2}$$

$$\text{Precision} = \frac{TP}{TP + FP} \tag{3}$$

$$\text{Specificity} = \frac{TN}{TN + FP} \tag{4}$$

$$\text{Recall} = \text{Sensitivity} = \frac{TP}{TP + FN} \tag{5}$$

$$F1\text{-score} = \frac{2 \times \text{Precision} \times \text{Recall}}{\text{Precision} + \text{Recall}} \tag{6}$$

$$\text{IoU} = \frac{TP}{TP + FP + FN} \tag{7}$$

$$\text{ROC} = \frac{\sqrt{\text{Sensitivity}^2 + \text{Specificity}^2}}{\sqrt{2}} \tag{8}$$

$$\text{Youden Index} = \text{Specificity} + \text{Sensitivity} - 100\% \tag{9}$$

$$\text{NPV} = \frac{TN}{TN + FN} \tag{10}$$

$$\text{WSM} = \frac{1}{10} \times (\text{Accuracy} + \text{Balanced Accuracy} + \text{Precision} + \text{Specificity} + \text{Recall} + \text{F1} + \text{IoU} + \text{ROC} + \text{Youden Index} + \text{NPV}) \tag{11}$$

## The suggested algorithm pseudocode

Algorithm 1 summarizes the different phases for fitting and testing a model. It is summarized graphically in Fig. 5.

**Algorithm 1**  The algorithm pseudocode for fitting and testing a model.

```
// Load the target dataset (one of the different created datasets in Subsection 3).
```
1 *tweets*, *labels* = **LoadDataset**()
```
// Create a tokenizer and fit the tweets on it.
```
2 *tokenizer* = **CreateFitTokenizer()**

3 *tokens* = *tokenizer. fit*(*tweets*)
```
// Convert the tweets to sequences.
```
4 *sequences* = **ConvertToSequences**(*tokens*)
```
// Pad the sequences with a predefined max length, post padding, and post truncating.
```
5 *paddedSequences* = **PadAndTruncate**(*sequences*)
```
// Convert the labels to categorical labels.
```
6 *catLabels* = **LabelsToCategoricalLabels**(*labels*)
```
// Create the model and initialize it. The model can be one of the model discussed in
```
   Table 7
7 *model* = **CreateModel**()
```
// Split the dataset into training, testing, and validation with a predefined train-to-
   test split percentage.
```
8 *trainX*, *trainY*, *testX*, *testY*, *valX*, *valY* = **SplitDataset**(*paddedSequences, catLabels*)
```
// Train and fit the created model on the datasets for a predefined epochs.
```
9 *model.fit*(*trainX, trainY, valX, valY*)
```
// Test the trained model using the testing subset and report the different performance
   metrics.
```
10 *metrics* = *model.evaluate*(*testX, testY*)

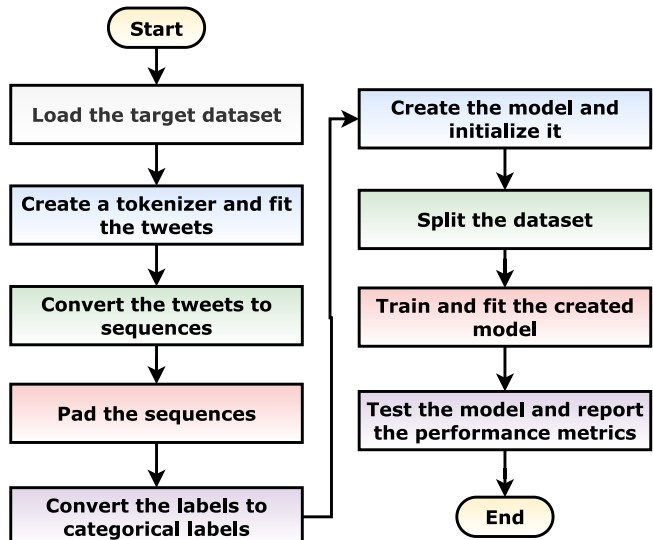

**Figure 5**  Graphical flowchart summarization of the suggested algorithm.

**Table 8 The experiments configurations.**

| Configuration | Specifications |
| --- | --- |
| Datasets | Subsection 3 |
| Sequence Max Length | 100 |
| No. of Epochs | 25 |
| Batch size | 32 |
| Train-to-test Size | 85% to 15% |
| Models | Table 7 |
| No. of USE Models | 3 |
| No. of BERT Models | 5 |
| Scripting Language | Python |
| Working Environment | Google Colab (*i.e.*, Intel(R) Xeon(R) CPU @ 2.00 GHz, Tesla T4 16 GB GPU, CUDA v.11.2, and 12 GB RAM) |

# EXPERIMENTS AND DISCUSSIONS

This section presents the experiments and the discussions of the reported results. The section is partitioned into two sections (1) experiments applied using the USE models and (2) experiments applied using the BERT models. Table 8 summarizes the common configurations of all experiments.

## The USE experiments

The current subsection discussed the applied experiments concerning the four USE flavors. Table 9 shows the loss and confusion matrix results of the four USE flavors (Table 7) on the six datasets (as listed in Subsection 3). From Table 9, the different discussed performance metrics can be calculated. They are reported in Table 10. From Table 9, it shows that the lowest loss of the different USE models is (1) 0.431 which is reported by "USE-MQA" using the "Before Normalization" dataset, (2) 0.473 which is reported by "USE-MQA" using the "After Normalization" dataset, (3) 0.526 which is reported by "USE-MQA" using the "Stemmed" dataset, (4) 0.477 which is reported by "USE-MQA" using the "Lemmatized" dataset, (5) 0.570 which is reported by "USE-M" using the "Stemmed and Filtered" dataset, and (6) 0.539 which is reported by "USE-MQA" using the "Lemmatized and Filtered" dataset. From Table 10, it shows that the highest WSM value of the different USE models is (1) 80.20% using the "Before Normalization" dataset, (2) 74.35% using the "After Normalization" dataset, (3) 77.47% using the "Stemmed" dataset, (4) 73.13% using the "Lemmatized" dataset, (5) 71.39% using the "Stemmed and Filtered" dataset, and (6) 70.59% using the "Lemmatized and Filtered" dataset. The best overall WSM value is 80.20%, produced by the "USE-MQA" model and "Before Normalization" dataset. The lowest overall WSM value is 61.06%, produced by the "USE-MQA" model and "Stemmed and Filtered" dataset. The average WSM value is 71.42%. Table 11 summarizes the WSM metrics and presents the maximum values of each row and column. Also, they are summarized graphically in Fig. 6.

**Table 9 The loss and confusion matrix results of the four USE flavors on the six datasets.**

| Keyword | Dataset | Loss | TP | TN | FP | FN |
|---------|---------|------|-----|-----|-----|-----|
| USE-M (Tuned) | Before Normalization | 0.718 | 109 | 136 | 25 | 35 |
| USE-MQA (Tuned) | Before Normalization | 0.442 | 116 | 136 | 25 | 28 |
| USE-M | Before Normalization | 0.444 | 123 | 127 | 34 | 21 |
| USE-MQA | Before Normalization | 0.431 | 111 | 144 | 17 | 33 |
| USE-M (Tuned) | After Normalization | 1.063 | 111 | 118 | 43 | 33 |
| USE-MQA (Tuned) | After Normalization | 1.338 | 114 | 118 | 43 | 30 |
| USE-M | After Normalization | 0.538 | 117 | 112 | 49 | 27 |
| USE-MQA | After Normalization | 0.473 | 115 | 123 | 38 | 29 |
| USE-M (Tuned) | Stemmed | 0.750 | 115 | 132 | 29 | 29 |
| USE-MQA (Tuned) | Stemmed | 1.590 | 103 | 126 | 35 | 41 |
| USE-M | Stemmed | 0.527 | 113 | 111 | 50 | 31 |
| USE-MQA | Stemmed | 0.526 | 94 | 130 | 31 | 50 |
| USE-M (Tuned) | Lemmatized | 0.483 | 90 | 142 | 19 | 54 |
| USE-MQA (Tuned) | Lemmatized | 1.344 | 102 | 123 | 38 | 42 |
| USE-M | Lemmatized | 0.523 | 113 | 114 | 47 | 31 |
| USE-MQA | Lemmatized | 0.477 | 112 | 123 | 38 | 32 |
| USE-M (Tuned) | Stemmed and Filtered | 1.039 | 106 | 125 | 36 | 38 |
| USE-MQA (Tuned) | Stemmed and Filtered | 1.189 | 93 | 120 | 41 | 51 |
| USE-M | Stemmed and Filtered | 0.570 | 100 | 115 | 46 | 44 |
| USE-MQA | Stemmed and Filtered | 0.598 | 68 | 139 | 22 | 76 |
| USE-M (Tuned) | Lemmatized and Filtered | 1.102 | 101 | 123 | 38 | 43 |
| USE-MQA (Tuned) | Lemmatized and Filtered | 0.877 | 103 | 123 | 38 | 41 |
| USE-M | Lemmatized and Filtered | 0.618 | 104 | 121 | 40 | 40 |
| USE-MQA | Lemmatized and Filtered | 0.539 | 114 | 113 | 48 | 30 |

   **The USE Experiments Remarks**: It can be conducted from Tables 9–11, and Fig. 6 that (1) the lemmatization is better than stemming concerning all non-tuned USE models without filtering and the "USE-M (Tuned)" model with filtering and the stemming is better than lemmatization otherwise, (2) the lemmatization without filtering is better than the lemmatization with filtering concerning all USE models unless the "USE-MQA (Tuned)" model, (3) the stemming without filtering is better than the stemming with filtering concerning all USE models, and (4) there are major differences between using the normalization and ignoring it where the ignorance is better.

## The BERT experiments

The current subsection discussed the applied experiments concerning the five BERT flavors. Table 12 shows the loss and confusion matrix results of the five BERT flavors (Table 7) on the six datasets (as listed in Subsection 3). The different discussed performance metrics can be calculated from Table 12. They are reported in Table 13. From Table 12, it shows that the lowest loss of the different BERT models is (1) 0.443 which is

**Table 10 The performance metrics of the four USE flavors on the six datasets.**

| Keyword | Dataset | Accuracy (%) | Balanced accuracy (%) | Precision (%) | Specificity (%) | Recall (%) | F1 (%) | IoU (%) | ROC (%) | Youden index (%) | NPV (%) | WSM (%) |
|---|---|---|---|---|---|---|---|---|---|---|---|---|
| USE-M (Tuned) | Before Normalization | 80.33 | 80.08 | 81.34 | 84.47 | 75.69 | 78.42 | 64.50 | 80.20 | 60.17 | 79.53 | 76.47 |
| USE-MQA (Tuned) | Before Normalization | 82.62 | 82.51 | 82.27 | 84.47 | 80.56 | 81.40 | 68.64 | 82.54 | 65.03 | 82.93 | 79.30 |
| USE-M | Before Normalization | 81.97 | 82.15 | 78.34 | 78.88 | 85.42 | 81.73 | 69.10 | 82.21 | 64.30 | 85.81 | 78.99 |
| USE-MQA | Before Normalization | 83.61 | 83.26 | 86.72 | 89.44 | 77.08 | 81.62 | 68.94 | 83.49 | 66.52 | 81.36 | 80.20 |
| USE-M (Tuned) | After Normalization | 75.08 | 75.19 | 72.08 | 73.29 | 77.08 | 74.50 | 59.36 | 75.21 | 50.38 | 78.15 | 71.03 |
| USE-MQA (Tuned) | After Normalization | 76.07 | 76.23 | 72.61 | 73.29 | 79.17 | 75.75 | 60.96 | 76.29 | 52.46 | 79.73 | 72.25 |
| USE-M | After Normalization | 75.08 | 75.41 | 70.48 | 69.57 | 81.25 | 75.48 | 60.62 | 75.63 | 50.82 | 80.58 | 71.49 |
| USE-MQA | After Normalization | 78.03 | 78.13 | 75.16 | 76.40 | 79.86 | 77.44 | 63.19 | 78.15 | 56.26 | 80.92 | 74.35 |
| USE-M (Tuned) | Stemmed | 80.98 | 80.92 | 79.86 | 81.99 | 79.86 | 79.86 | 66.47 | 80.93 | 61.85 | 81.99 | 77.47 |
| USE-MQA (Tuned) | Stemmed | 75.08 | 74.89 | 74.64 | 78.26 | 71.53 | 73.05 | 57.54 | 74.97 | 49.79 | 75.45 | 70.52 |
| USE-M | Stemmed | 73.44 | 73.71 | 69.33 | 68.94 | 78.47 | 73.62 | 58.25 | 73.86 | 47.42 | 78.17 | 69.52 |
| USE-MQA | Stemmed | 73.44 | 73.01 | 75.20 | 80.75 | 65.28 | 69.89 | 53.71 | 73.42 | 46.02 | 72.22 | 68.29 |
| USE-M (Tuned) | Lemmatized | 76.07 | 75.35 | 82.57 | 88.20 | 62.50 | 71.15 | 55.21 | 76.44 | 50.70 | 72.45 | 71.06 |
| USE-MQA (Tuned) | Lemmatized | 73.77 | 73.62 | 72.86 | 76.40 | 70.83 | 71.83 | 56.04 | 73.67 | 47.23 | 74.55 | 69.08 |
| USE-M | Lemmatized | 74.43 | 74.64 | 70.63 | 70.81 | 78.47 | 74.34 | 59.16 | 74.74 | 49.28 | 78.62 | 70.51 |
| USE-MQA | Lemmatized | 77.05 | 77.09 | 74.67 | 76.40 | 77.78 | 76.19 | 61.54 | 77.09 | 54.18 | 79.35 | 73.13 |
| USE-M (Tuned) | Stemmed and Filtered | 75.74 | 75.63 | 74.65 | 77.64 | 73.61 | 74.13 | 58.89 | 75.65 | 51.25 | 76.69 | 71.39 |
| USE-MQA (Tuned) | Stemmed and Filtered | 69.84 | 69.56 | 69.40 | 74.53 | 64.58 | 66.91 | 50.27 | 69.74 | 39.12 | 70.18 | 64.41 |
| USE-M | Stemmed and Filtered | 70.49 | 70.44 | 68.49 | 71.43 | 69.44 | 68.97 | 52.63 | 70.44 | 40.87 | 72.33 | 65.55 |
| USE-MQA | Stemmed and Filtered | 67.87 | 66.78 | 75.56 | 86.34 | 47.22 | 58.12 | 40.96 | 69.58 | 33.56 | 64.65 | 61.06 |
| USE-M (Tuned) | Lemmatized and Filtered | 73.44 | 73.27 | 72.66 | 76.40 | 70.14 | 71.38 | 55.49 | 73.33 | 46.54 | 74.10 | 68.67 |
| USE-MQA (Tuned) | Lemmatized and Filtered | 74.10 | 73.96 | 73.05 | 76.40 | 71.53 | 72.28 | 56.59 | 74.00 | 47.93 | 75.00 | 69.48 |
| USE-M | Lemmatized and Filtered | 73.77 | 73.69 | 72.22 | 75.16 | 72.22 | 72.22 | 56.52 | 73.70 | 47.38 | 75.16 | 69.20 |
| USE-MQA | Lemmatized and Filtered | 74.43 | 74.68 | 70.37 | 70.19 | 79.17 | 74.51 | 59.38 | 74.81 | 49.35 | 79.02 | 70.59 |

reported by "BERT-BV2" using the "Lemmatized" dataset, (2) 0.371, which is reported by "BERT-LV2" using the "Lemmatized and Filtered" dataset, (3) 0.384 which is reported by "AraElectra" using the "Before Normalization" dataset, (4) 0.330 which is reported by "BERT-BV02T" using the "Before Normalization" dataset, and (5) 0.404 which is reported by "BERT-LV02T" using the "After Normalization" dataset. From Table 13, it shows that

**Table 11 The WSM metrics of the four USE flavors on the six datasets.**

|  | Before normalization (%) | After normalization (%) | Stemmed (%) | Lemmatized (%) | Stemmed and filtered (%) | Lemmatized and filtered (%) | MAX (%) |
|---|---|---|---|---|---|---|---|
| USE-M (Tuned) | 76.47 | 71.03 | 77.47 | 71.06 | 71.39 | 68.67 | 77.47 |
| USE-MQA (Tuned) | 79.30 | 72.25 | 70.52 | 69.08 | 64.41 | 69.48 | 79.30 |
| USE-M | 78.99 | 71.49 | 69.52 | 70.51 | 65.55 | 69.20 | 78.99 |
| USE-MQA | **80.20** | 74.35 | 68.29 | 73.13 | 61.06 | 70.59 | **80.20** |
| MAX | **80.20** | 74.35 | 77.47 | 73.13 | 71.39 | 70.59 | **80.20** |

Note:
The bold results reflect the best reported results.

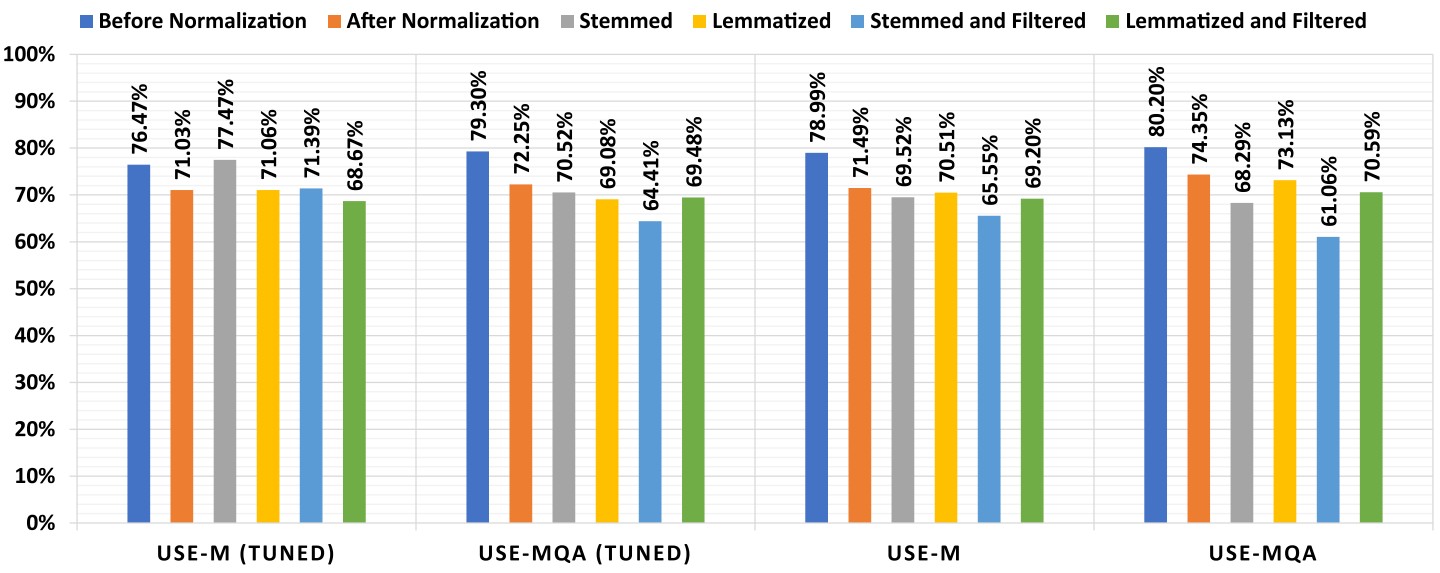

**Figure 6 Graphical summarization of the WSM metrics of the four USE flavors on the six datasets.**

the highest WSM value of the different BERT models is (1) 95.26% using the "Before Normalization" dataset, (2) 94.67% using the "After Normalization" dataset, (3) 92.96% using the "Stemmed" dataset, (4) 94.40%, using the "Lemmatized" dataset, (5) 92.91% using the "Stemmed and Filtered" dataset, and (6) 93.49% using the "Lemmatized and Filtered" dataset. The best overall WSM value is 95.26%, produced by the "BERT-BV02T" model and the "Before Normalization" dataset. The lowest overall WSM value is 38.30%, produced by the "BERT-LV2" model and "Stemmed" dataset. The average WSM value is 79.33%. Table 14 summarizes the WSM metrics and presents the maximum values of each row and column. Also, they are summarized graphically in Fig. 7.

**The BERT Experiments Remarks**: It can be conduced from Tables 12–14, and Fig. 7 that (1) the "BERT-LV2" and "BERT-LV02T" are not performing well, (2) the lemmatization is better than stemming concerning all BERT models without filtering and

**Table 12  The loss and confusion matrix results of the five BERT flavors on the six datasets.**

| Keyword | Dataset | Loss | TP | TN | FP | FN |
|---|---|---|---|---|---|---|
| BERT-BV2 | Before Normalization | 0.478 | 927 | 990 | 84 | 29 |
| BERT-BV2 | After Normalization | 0.464 | 929 | 993 | 81 | 27 |
| BERT-BV2 | Stemmed | 0.544 | 922 | 987 | 87 | 34 |
| BERT-BV2 | Lemmatized | 0.443 | 922 | 1,008 | 66 | 34 |
| BERT-BV2 | Stemmed and Filtered | 0.560 | 923 | 984 | 90 | 33 |
| BERT-BV2 | Lemmatized and Filtered | 0.460 | 921 | 998 | 76 | 35 |
| BERT-LV2 | Before Normalization | 0.692 | 0 | 1,074 | 0 | 956 |
| BERT-LV2 | After Normalization | 0.691 | 0 | 1,074 | 0 | 956 |
| BERT-LV2 | Stemmed | 0.664 | 24 | 1,050 | 24 | 932 |
| BERT-LV2 | Lemmatized | 0.694 | 0 | 1,074 | 0 | 956 |
| BERT-LV2 | Stemmed and Filtered | 0.692 | 0 | 1,074 | 0 | 956 |
| BERT-LV2 | Lemmatized and Filtered | 0.371 | 904 | 1,005 | 69 | 52 |
| AraElectra | Before Normalization | 0.384 | 921 | 1,003 | 71 | 35 |
| AraElectra | After Normalization | 0.387 | 909 | 1,014 | 60 | 47 |
| AraElectra | Stemmed | 0.411 | 903 | 995 | 79 | 53 |
| AraElectra | Lemmatized | 0.403 | 898 | 1,006 | 68 | 58 |
| AraElectra | Stemmed and Filtered | 0.473 | 906 | 989 | 85 | 50 |
| AraElectra | Lemmatized and Filtered | 0.399 | 910 | 985 | 89 | 46 |
| BERT-BV02T | Before Normalization | 0.330 | 926 | 1,024 | 50 | 30 |
| BERT-BV02T | After Normalization | 0.409 | 931 | 1,005 | 69 | 25 |
| BERT-BV02T | Stemmed | 0.543 | 907 | 1,004 | 70 | 49 |
| BERT-BV02T | Lemmatized | 0.418 | 921 | 1,009 | 65 | 35 |
| BERT-BV02T | Stemmed and Filtered | 0.557 | 896 | 1,002 | 72 | 60 |
| BERT-BV02T | Lemmatized and Filtered | 0.495 | 919 | 992 | 82 | 37 |
| BERT-LV02T | Before Normalization | 0.692 | 0 | 1,074 | 0 | 956 |
| BERT-LV02T | After Normalization | 0.404 | 932 | 1,007 | 67 | 24 |
| BERT-LV02T | Stemmed | 0.691 | 0 | 1,074 | 0 | 956 |
| BERT-LV02T | Lemmatized | 0.468 | 923 | 1,012 | 62 | 33 |
| BERT-LV02T | Stemmed and Filtered | 0.538 | 908 | 1,002 | 72 | 48 |
| BERT-LV02T | Lemmatized and Filtered | 0.657 | 24 | 1,071 | 3 | 932 |

all BERT model with filtering unless the "BERT-LV02T" model, (3) the lemmatization without filtering is better than the lemmatization with filtering concerning all BERT models unless the "BERT-LV2" model, (4) the stemming without filtering is better than the stemming with filtering concerning all BERT models unless the "BERT-LV2" and "BERT-LV02T" models, and (5) there are no major differences between using the normalization or ignoring it.

**Table 13 The performance metrics of the five BERT flavors on the six datasets.**

| Keyword | Dataset | Accuracy (%) | Balanced accuracy (%) | Precision (%) | Specificity (%) | Recall (%) | F1 (%) | IoU (%) | ROC (%) | Youden index (%) | NPV (%) | WSM (%) |
|---|---|---|---|---|---|---|---|---|---|---|---|---|
| BERT-BV2 | Before Normalization | 94.43 | 94.57 | 91.69 | 92.18 | 96.97 | 94.26 | 89.13 | 94.60 | 89.15 | 97.15 | 93.41 |
| BERT-BV2 | After Normalization | 94.68 | 94.82 | 91.98 | 92.46 | 97.18 | 94.51 | 89.59 | 94.85 | 89.63 | 97.35 | 93.70 |
| BERT-BV2 | Stemmed | 94.04 | 94.17 | 91.38 | 91.90 | 96.44 | 93.84 | 88.40 | 94.20 | 88.34 | 96.67 | 92.94 |
| BERT-BV2 | Lemmatized | 95.07 | 95.15 | 93.32 | 93.85 | 96.44 | 94.86 | 90.22 | 95.16 | 90.30 | 96.74 | 94.11 |
| BERT-BV2 | Stemmed and Filtered | 93.94 | 94.08 | 91.12 | 91.62 | 96.55 | 93.75 | 88.24 | 94.12 | 88.17 | 96.76 | 92.83 |
| BERT-BV2 | Lemmatized and Filtered | 94.53 | 94.63 | 92.38 | 92.92 | 96.34 | 94.32 | 89.24 | 94.65 | 89.26 | 96.61 | 93.49 |
| BERT-LV2 | Before Normalization | 52.91 | 50 | N/A | 100 | 0 | N/A | 0 | 70.71 | 0 | 52.91 | 40.82 |
| BERT-LV2 | After Normalization | 52.91 | 50 | N/A | 100 | 0 | N/A | 0 | 70.71 | 0 | 52.91 | 40.82 |
| BERT-LV2 | Stemmed | 52.91 | 50.14 | 50 | 97.77 | 2.51 | 4.78 | 2.45 | 69.15 | 0.28 | 52.98 | 38.30 |
| BERT-LV2 | Lemmatized | 52.91 | 50 | N/A | 100 | 0 | N/A | 0 | 70.71 | 0 | 52.91 | 40.82 |
| BERT-LV2 | Stemmed and Filtered | 52.91 | 50 | N/A | 100 | 0 | N/A | 0 | 70.71 | 0 | 52.91 | 40.82 |
| BERT-LV2 | Lemmatized and Filtered | 94.04 | 94.07 | 92.91 | 93.58 | 94.56 | 93.73 | 88.20 | 94.07 | 88.14 | 95.08 | 92.84 |
| AraElectra | Before Normalization | 94.78 | 94.86 | 92.84 | 93.39 | 96.34 | 94.56 | 89.68 | 94.88 | 89.73 | 96.63 | 93.77 |
| AraElectra | After Normalization | 94.73 | 94.75 | 93.81 | 94.41 | 95.08 | 94.44 | 89.47 | 94.75 | 89.50 | 95.57 | 93.65 |
| AraElectra | Stemmed | 93.50 | 93.55 | 91.96 | 92.64 | 94.46 | 93.19 | 87.25 | 93.55 | 87.10 | 94.94 | 92.21 |
| AraElectra | Lemmatized | 93.79 | 93.80 | 92.96 | 93.67 | 93.93 | 93.44 | 87.70 | 93.80 | 87.60 | 94.55 | 92.52 |
| AraElectra | Stemmed and Filtered | 93.35 | 93.43 | 91.42 | 92.09 | 94.77 | 93.07 | 87.03 | 93.44 | 86.86 | 95.19 | 92.06 |
| AraElectra | Lemmatized and Filtered | 93.35 | 93.45 | 91.09 | 91.71 | 95.19 | 93.09 | 87.08 | 93.47 | 86.90 | 95.54 | 92.09 |
| BERT-BV02T | Before Normalization | 96.06 | 96.10 | 94.88 | 95.34 | 96.86 | 95.86 | 92.05 | 96.11 | 92.21 | 97.15 | 95.26 |
| BERT-BV02T | After Normalization | 95.37 | 95.48 | 93.10 | 93.58 | 97.38 | 95.19 | 90.83 | 95.50 | 90.96 | 97.57 | 94.50 |
| BERT-BV02T | Stemmed | 94.14 | 94.18 | 92.84 | 93.48 | 94.87 | 93.84 | 88.40 | 94.18 | 88.36 | 95.35 | 92.96 |
| BERT-BV02T | Lemmatized | 95.07 | 95.14 | 93.41 | 93.95 | 96.34 | 94.85 | 90.21 | 95.15 | 90.29 | 96.65 | 94.11 |
| BERT-BV02T | Stemmed and Filtered | 93.50 | 93.51 | 92.56 | 93.30 | 93.72 | 93.14 | 87.16 | 93.51 | 87.02 | 94.35 | 92.18 |
| BERT-BV02T | Lemmatized and Filtered | 94.14 | 94.25 | 91.81 | 92.36 | 96.13 | 93.92 | 88.54 | 94.27 | 88.49 | 96.40 | 93.03 |
| BERT-LV02T | Before Normalization | 52.91 | 50 | N/A | 100 | 0 | N/A | 0 | 70.71 | 0 | 52.91 | 40.82 |
| BERT-LV02T | After Normalization | 95.52 | 95.63 | 93.29 | 93.76 | 97.49 | 95.35 | 91.10 | 95.64 | 91.25 | 97.67 | 94.67 |
| BERT-LV02T | Stemmed | 52.91 | 50 | N/A | 100 | 0 | N/A | 0 | 70.71 | 0 | 52.91 | 40.82 |
| BERT-LV02T | Lemmatized | 95.32 | 95.39 | 93.71 | 94.23 | 96.55 | 95.11 | 90.67 | 95.39 | 90.78 | 96.84 | 94.40 |
| BERT-LV02T | Stemmed and Filtered | 94.09 | 94.14 | 92.65 | 93.30 | 94.98 | 93.80 | 88.33 | 94.14 | 88.28 | 95.43 | 92.91 |
| BERT-LV02T | Lemmatized and Filtered | 53.94 | 51.12 | 88.89 | 99.72 | 2.51 | 4.88 | 2.50 | 70.54 | 2.23 | 53.47 | 42.98 |

**Table 14 The WSM metrics of the five BERT flavors on the six datasets.**

| | Before normalization (%) | After normalization (%) | Stemmed (%) | Lemmatized (%) | Stemmed and filtered (%) | Lemmatized and filtered (%) | MAX (%) |
|---|---|---|---|---|---|---|---|
| BERT-BV2 | 93.41 | 93.70 | 92.94 | 94.11 | 92.83 | 93.49 | 94.11 |
| BERT-LV2 | 40.82 | 40.82 | 38.30 | 40.82 | 40.82 | 92.84 | 92.84 |
| AraElectra | 93.77 | 93.65 | 92.21 | 92.52 | 92.06 | 92.09 | 93.77 |
| BERT-BV02T | 95.26 | 94.50 | 92.96 | 94.11 | 92.18 | 93.03 | 95.26 |
| BERT-LV02T | 40.82 | 94.67 | 40.82 | 94.40 | 92.91 | 42.98 | 94.67 |
| MAX | 95.26 | 94.67 | 92.96 | 94.40 | 92.91 | 93.49 | 95.26 |

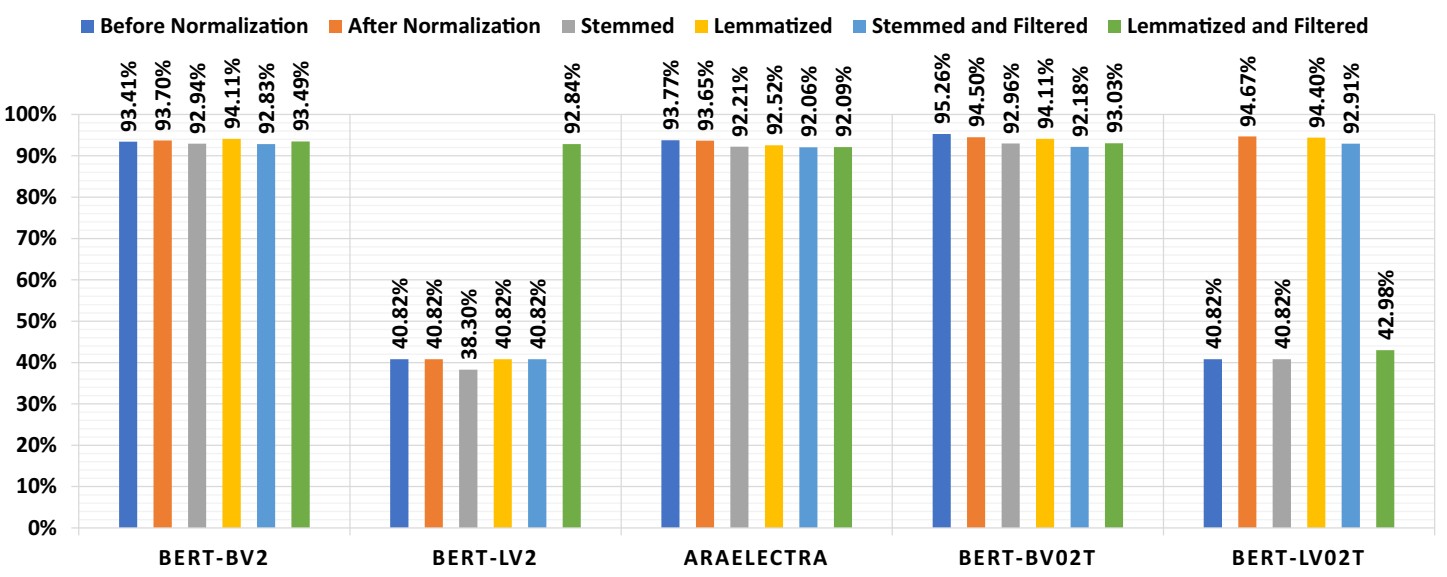

**Figure 7 Graphical summarization of the WSM metrics of the five BERT flavors on the six datasets.**

## CONCLUSIONS

Various mental disorders such as MDD (or depression) affect millions of individuals worldwide. These disorders affect their daily lives. More than 300 million people worldwide are affected, and the number is growing daily. For example, it is estimated that 800K teens commit suicide yearly, making it the second leading cause of death in youth. Social networking sites such as Twitter and Facebook allow individuals to share their feelings and opinions on various subjects. The process also leads to the generation of social data, providing important insight into people's mental health and daily activities. In studying and analyzing user data, researchers can discover how mental health influences human behavior and how they can assist users in recovering. Researchers have used social media data to track depression and other mental illnesses. Social media data detect depressive symptoms discreetly before they progress to more severe stages. Consequently,

prevention and therapy strategies can be recommended to patients early. NLP is used to analyze spoken and written language. Languages like Arabic and English can be processed using NLP. SA identifies positive and negative aspects of a text, speech, or social media post. This method can analyze a person's depression level based on social media posts and negative sentiment ratings. The present study presents a complete framework for performing text classification (*i.e.*, tweets). The framework can be applied in six phases beginning with the data acquisition phase and transferring to classification through two cascading phases. The trained and optimized model classifies the Arabic tweets as "Suicide" or "Normal" using the trained and optimized model. The work also provides a preprocessing algorithm for Arabic tweets. Five different annotators have annotated a recently scraped Arabic tweets dataset, while the system performance was assessed using different performance metrics. General insights were obtained from the WSM. A USE model and a BERT model in Arabic were investigated. USE models have the best WSM at 80.20%, and Arabic BERT models have the best WSM at 95.26%. Different models can test a wider range of scraped data in future studies. The work can be extended by applying automatic learning for Arabic to accept and predict future tweets.

### Funding
This work was supported by Princess Nourah bint Abdulrahman University, Researchers Supporting Project number (PNURSP2022R293), Princess Nourah bint Abdulrahman University, Riyadh, Saudi Arabia. The funders had no role in study design, data collection and analysis, decision to publish, or preparation of the manuscript.

### Grant Disclosures
The following grant information was disclosed by the authors:
Princess Nourah bint Abdulrahman University, Researchers Supporting Project number: NURSP2022R293.

### Competing Interests
The authors declare that they have no competing interests.

### Author Contributions
- Nadiah A. Baghdadi analyzed the data, prepared figures and/or tables, authored or reviewed drafts of the article, and approved the final draft.
- Amer Malki performed the experiments, analyzed the data, prepared figures and/or tables, authored or reviewed drafts of the article, and approved the final draft.
- Hossam Magdy Balaha conceived and designed the experiments, performed the experiments, analyzed the data, performed the computation work, authored or reviewed drafts of the article, and approved the final draft.
- Yousry AbdulAzeem conceived and designed the experiments, performed the experiments, analyzed the data, performed the computation work, authored or reviewed drafts of the article, and approved the final draft.

- Mahmoud Badawy conceived and designed the experiments, performed the experiments, analyzed the data, prepared figures and/or tables, authored or reviewed drafts of the article, and approved the final draft.
- Mostafa Elhosseini conceived and designed the experiments, performed the experiments, analyzed the data, performed the computation work, authored or reviewed drafts of the article, and approved the final draft.

## Data Availability

Data is available at GitHub: https://github.com/HossamBalaha/An-Optimized-Deep-Learning-Approach-for-Suicide-Detection-through-Arabic-Tweets.

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
