# Peer review of "An optimized deep learning approach for suicide detection through Arabic tweets"

_PeerJ Computer Science, doi:10.7717/peerj-cs.1070_

## Round 0.1 · original submission · Major Revisions

Your manuscript has been reviewed by the experts in the field and they are suggesting a lots of improvements, therefore you are requested to make major changes in the paper and resubmit for consideration.

Reviewer 1 ·

Basic reporting

The authors in this work presented a Deep-learning based classification model that utilizes Arabic tweet data to categorize users as potential suicides. The paper is well written and structured. One of my main concerns is the novelty of the work. The authors should carefully address all the following concerns, especially the comments 4—7:

Experimental design

Please see the additional comments section of the review.

Validity of the findings

Please see the additional comments section of the review.

Additional comments

1. In the abstract, authors should provide a highlight (in 2 to 3 sentences) about the existing landscape (i.e., available related work) and their limitations with a brief justification of the proposed work
2. Line 32-33 (introduction Section Page 1): The statement “More than 300 million individuals are impacted globally, accounting for 4.4% of the global population (WHO), and the number is growing every day” should have a reference.
a. In general, every indicated result or data related to some study should be properly referenced.
b. Example: (line 40 Page 1) “Every year, about 800,000 individuals commit suicide.” Without any reference is not acceptable.
3. Page 2, lines: 90-91: the statement ” However, the traditional ML technique’s performance was restricted as the number of correlations rose dramatically due to the rise in data amount Rao et al. (2020)”. The authors should provide the rationale in terms of the amount of data with certain quantification (i.e., how much it should be?) for the nature of data it is suitable for.. etc
a. A motivational example related to the justification or rationale of using DL instead of traditional ML techniques should be provided.
4. My main concerns are the novelty of the work. In the introduction section, the authors must mention the existing tweet classification techniques (both MPO and DL based) and should JUSTIFY how their approach is novel. Without justifying novelty, In the current form, this work looks like just another study and lacks the merit of a research paper with respect to the novelty (should be based on some novel aspect in the knowledge domain).
5. Related work section should be extended to add more techniques and the recent ones from the year 2021 and year 2022.
6. The comparison of the related work technique is missing. Add a comparison table at the end of the related work section and especially highlight the potential demerits of the existing approach in that table.
7. Based on the related work comparison table (as suggested in point 6), add a research gap statement highlighting the potential shortcoming of the existing approaches so that the proposed solution could be justifiable and be considered novel.
8. “Twint tool”, add a reference to this python tool.
9. The authors collected tweet data using Arabic-related words equivalent to “suicide”. Which other keywords could also be helpful and were not used such as Bipolar disorder, self-killing, lethal, etc. Please add relevant discussion.
10. Equations (1 to 11) should have references if taken from the literature.
11. Algorithm 1 (on page 9) is just a sequence of function calls and does not show the flow of the relevant data items. The appropriate data structures, parameters, and return values should be made part of Algorithm 1 and the mentioned function calls.
12. Results discussion is too brief. The table and figures related to results should have a more relevant discussion special the reasons or rationale for the attainment of better and poorer performances.
13. Table 13 and Figure 6 present redundant result data. Use wither table or figure to depict these results.

Reviewer 2 ·

Basic reporting

This paper presents an optimized deep learning approach for suicide detection through Arabic tweets. The authors used the data from Twitter of Arabic tweets to classify whether a tweet is categorized as a normal or a suicide case. Some of my questions are

You used the five annotators to annotate the data, who were they? did they annotate the data automatically or manually?
Which State-of-the-art performance metrics are reported, please mention those (names) in abstract
An evaluation discussion is needed in the abstract.

Line 109: What follows is a summary of the rest of the paper: In Section 2, the related studies are presented. 110 Section 3 discusses section numbers are missing throughout the manuscript
A very relevant and state of the art study published on similar topic may also be discussed in the related work section
https://www.sciencedirect.com/science/article/abs/pii/S0736585320300046
many grammatical mistakes could be corrected . e.g. line 164-165 says
Table 2 show 5 samples from the retrieved records.
It could be
Table 2 shows five samples from the retrieved records.
Line 187, you may use some appropriate word in replacement of appliance
Line 192, there is no sixth row in table 2, (e.g., the sixth row in Table 2).
The authors mentioned the manual annotation, is it not a tediuous task for large data sets ?, chould you find an automatic way of annotations? Because there are some tools available.
How did you performed Stemming and Lemmatization , stop words removal on Arabic , again manually ? or through some automated tools ?
Regarding table 5 the model USE and BERT are for Arabic language as well ? or did you custimazied them?
Algorithm 1 could be written in more professional way
Line 256, The current section presents could be more appropriately written as This section……
It is recommended to add a comparisons and evaluation metrics before the conclusion section

Experimental design

This paper presents an optimized deep learning approach for suicide detection through Arabic tweets. The authors used the data from Twitter of Arabic tweets to classify whether a tweet is categorized as a normal or a suicide case. Some of my questions are

You used the five annotators to annotate the data, who were they? did they annotate the data automatically or manually?
Which State-of-the-art performance metrics are reported, please mention those (names) in abstract
An evaluation discussion is needed in the abstract.

Line 109: What follows is a summary of the rest of the paper: In Section 2, the related studies are presented. 110 Section 3 discusses section numbers are missing throughout the manuscript
A very relevant and state of the art study published on similar topic may also be discussed in the related work section
https://www.sciencedirect.com/science/article/abs/pii/S0736585320300046
many grammatical mistakes could be corrected . e.g. line 164-165 says
Table 2 show 5 samples from the retrieved records.
It could be
Table 2 shows five samples from the retrieved records.
Line 187, you may use some appropriate word in replacement of appliance
Line 192, there is no sixth row in table 2, (e.g., the sixth row in Table 2).
The authors mentioned the manual annotation, is it not a tediuous task for large data sets ?, chould you find an automatic way of annotations? Because there are some tools available.
How did you performed Stemming and Lemmatization , stop words removal on Arabic , again manually ? or through some automated tools ?
Regarding table 5 the model USE and BERT are for Arabic language as well ? or did you custimazied them?
Algorithm 1 could be written in more professional way
Line 256, The current section presents could be more appropriately written as This section……
It is recommended to add a comparisons and evaluation metrics before the conclusion section

Validity of the findings

This paper presents an optimized deep learning approach for suicide detection through Arabic tweets. The authors used the data from Twitter of Arabic tweets to classify whether a tweet is categorized as a normal or a suicide case. Some of my questions are

You used the five annotators to annotate the data, who were they? did they annotate the data automatically or manually?
Which State-of-the-art performance metrics are reported, please mention those (names) in abstract
An evaluation discussion is needed in the abstract.

Line 109: What follows is a summary of the rest of the paper: In Section 2, the related studies are presented. 110 Section 3 discusses section numbers are missing throughout the manuscript
A very relevant and state of the art study published on similar topic may also be discussed in the related work section
https://www.sciencedirect.com/science/article/abs/pii/S0736585320300046
many grammatical mistakes could be corrected . e.g. line 164-165 says
Table 2 show 5 samples from the retrieved records.
It could be
Table 2 shows five samples from the retrieved records.
Line 187, you may use some appropriate word in replacement of appliance
Line 192, there is no sixth row in table 2, (e.g., the sixth row in Table 2).
The authors mentioned the manual annotation, is it not a tediuous task for large data sets ?, chould you find an automatic way of annotations? Because there are some tools available.
How did you performed Stemming and Lemmatization , stop words removal on Arabic , again manually ? or through some automated tools ?
Regarding table 5 the model USE and BERT are for Arabic language as well ? or did you custimazied them?
Algorithm 1 could be written in more professional way
Line 256, The current section presents could be more appropriately written as This section……
It is recommended to add a comparisons and evaluation metrics before the conclusion section

Additional comments

This paper presents an optimized deep learning approach for suicide detection through Arabic tweets. The authors used the data from Twitter of Arabic tweets to classify whether a tweet is categorized as a normal or a suicide case. Some of my questions are

You used the five annotators to annotate the data, who were they? did they annotate the data automatically or manually?
Which State-of-the-art performance metrics are reported, please mention those (names) in abstract
An evaluation discussion is needed in the abstract.

Line 109: What follows is a summary of the rest of the paper: In Section 2, the related studies are presented. 110 Section 3 discusses section numbers are missing throughout the manuscript
A very relevant and state of the art study published on similar topic may also be discussed in the related work section
https://www.sciencedirect.com/science/article/abs/pii/S0736585320300046
many grammatical mistakes could be corrected . e.g. line 164-165 says
Table 2 show 5 samples from the retrieved records.
It could be
Table 2 shows five samples from the retrieved records.
Line 187, you may use some appropriate word in replacement of appliance
Line 192, there is no sixth row in table 2, (e.g., the sixth row in Table 2).
The authors mentioned the manual annotation, is it not a tediuous task for large data sets ?, chould you find an automatic way of annotations? Because there are some tools available.
How did you performed Stemming and Lemmatization , stop words removal on Arabic , again manually ? or through some automated tools ?
Regarding table 5 the model USE and BERT are for Arabic language as well ? or did you custimazied them?
Algorithm 1 could be written in more professional way
Line 256, The current section presents could be more appropriately written as This section……
It is recommended to add a comparisons and evaluation metrics before the conclusion section

---

## Round 0.2 · accepted · Accept

Thank you for your submission, and good luck for your future research.

Reviewer 1 ·

Basic reporting

The revisions are satisfactory. No further comments.

Experimental design

The revisions are satisfactory. No further comments.

Validity of the findings

The revisions are satisfactory. No further comments.

Additional comments

The revisions are satisfactory. No further comments.

Reviewer 2 ·

Basic reporting

the comments are addressed

Experimental design

the comments are addressed

Validity of the findings

the comments are addressed

Additional comments

the comments are addressed